# Exploring the effects of IFN-τ on LPS-induced endometritis in cows based on transcriptomics

**Pan Liu**[1,2], **Yaofeng Zhang**[1,2], **Xu Chen**[1,2], **Lijun Tang**[1,2,3], **Wenxiu Xu**[1,2], **Guigui Wang**[1,2], **Duoqi Zhang**[1,2], **Junfeng Liu**[1,2]*

**1** College of Animal Science and Technology, Tarim University, Alar, Xinjiang, People's Republic of China, **2** Tarim Animal Disease Diagnosis and Prevention and Control Engineering Laboratory, Xinjiang Production and Construction Corps, Aral, Xinjiang, People's Republic of China, **3** Sichuan Vocational and Technical College, Suining, Sichuan, People's Republic of China

* ljfdky@126.com

## Abstract

Endometritis in dairy cows is a common reproductive disease that severely affects reproductive performance and milk production, resulting in significant economic losses. Lipopolysaccharide (LPS) as a PAMP capable of inducing inflammatory responses in the endometrium through the NF-κB pathway. Interferon-tau (IFN-τ) is a type I interferon with significant anti-inflammatory properties. Currently, transcriptomics sequencing technology has gradually become an attractive tool for studying such diseases. This study established an inflammatory model of bovine endometrial cells (BENDs) using LPS induction and employed RNA-seq technology to investigate the expression profiles of mRNAs in BENDs from the Control group (C), the LPS-treated group (L), and the IFN-τ+LPS-treated group (F). The results showed that there were 109, 1109, and 962 Differentially Expressed mRNAs (DEmRNAs) in the C vs L, C vs F, and L vs F. Gene Ontology (GO) and Kyoto Encyclopedia of Genes and Genomes (KEGG) enrichment analysis showed that these DEmRNAs were mainly involved in the regulation of host immune responses (e.g., NOD-like receptor signaling pathway, IL-17 signaling pathway and RIG-I-like receptor signaling pathway, NF-kappa B signaling pathway), signal transduction molecules and interactions (e.g., Cytokine-cytokine receptor interaction; Cell adhesion molecules), metabolic process (e.g., Glycosphingolipid biosynthesis-lacto and neolacto series) and Antigen processing and presentation, Complement and coagulation cascades, Th17 cell differentiation, etc. biological process. This study not only elucidates the molecular mechanisms by which BENDs respond to microbial invasion but also reveals the specific regulatory network through which IFN-τ exerts its anti-inflammatory effects via multiple synergistic pathways. It provides crucial theoretical support for the clinical application of interferon therapy in treating endometritis, demonstrating significant research value and promising applications.

**Data availability statement:** The "Transcriptomics Sequencing Data - mRNA-Seq sequencing data" in this study are available in the Sequence Read Archive (SRA) under project number PRJNA1337137 (https://www.ncbi.nlm.nih.gov/sra/PRJNA1337137).

**Funding:** This work was supported by the National Natural Science Foundation of China [grant numbers 32260905]. But the National Natural Science Foundation of China [grant number 32260905] had no role in study design, data collection and analysis, decision to publish, or preparation of the manuscript.

**Competing interests:** The authors declare that they have no known competing financial interests or personal relationships that could have appeared to influence the work reported in this paper. No potential conflict of interests relevant to this article was reported.

## 1. Introduction

Endometritis is a reproductive system disease in dairy cows caused by microorganisms invading the uterus through the open cervix during estrus or after calving [1–3]. This situation often leads to reproductive disorders and decreased milk production, resulting in significant economic losses [4]. *Escherichia coli* is the most common bacterial species isolated from the uterus postpartum and is associated with high concentrations of LPS in the uterine cavity of cows [5–7]. LPS is a well-known endotoxin and a major component of the cell wall of Gram-negative bacteria, playing a key role in the pathogenesis of bovine endometritis [8]. Over the years, several scholars have utilized LPS-induced inflammation models in bovine endometrial cells to identify numerous cellular and immune factors involved in the development of bovine endometritis, such as TNF-α, IL-1, IL-6, IL-8, IL-1β, and TLR4 [9–12]. These research findings have laid the foundation for elucidating the pathogenesis of bovine endometritis and for the diagnosis and prevention of the disease.

The transcriptome refers to the collection of all transcription products within cells under specific conditions, primarily including messenger RNA (mRNA), ribosomal RNA (rRNA), transfer RNA (tRNA), and non-coding RNA (e.g., microRNA, long non-coding RNA, small nucleolar RNA, circular RNA) [13]. In recent years, with the continuous development of transcriptomics and next-generation sequencing technologies, transcriptomics sequencing has become an increasingly attractive tool for identifying new diagnostic or therapeutic targets and treating tumors and infectious diseases by pinpointing genes or biological events of interest under specific conditions or disease states [14–17].

IFN-τ is a type I interferon that is specifically secreted by embryonic trophoblast cells during early pregnancy in ruminants. In addition to acting as a signal for maternal pregnancy recognition in cattle [18,19], it also has antiviral, antiproliferative, and immunomodulatory activities [20]. Initially recognized for its anti-luteolytic action through suppression of endometrial PGF2α. IFN-τ is now known to exert systemic effects beyond the uterus. It induces interferon-stimulated genes in endometrial and peripheral immune cells, shaping an immune environment conducive to embryo tolerance. By modulating NF-κB, signal transducer and activator of transcription 1, and interferon regulatory factors, IFN-τ downregulates pro-inflammatory cytokines such as tumor necrosis factor alpha and interferon gamma, while enhancing anti-inflammatory mediators including IL-10 and IL-4. This shift promotes a Thelper 2-dominant immune profile favorable for maternal–fetal tolerance [21]. Previous studies have shown that IFN-τ exerts significant anti-inflammatory effects in inflammatory diseases [22,23]. Still there are currently few studies exploring IFN-τ in LPS-induced endometritis in cows based on transcriptomics. Transcriptomics analysis is crucial for understanding the occurrence, development, and pathophysiological mechanisms of endometritis in dairy cows, which will facilitate the prevention and treatment of this disease.

This study established an inflammatory model of bovine endometrial epithelial cells induced by LPS. It utilized RNA-seq sequencing technology to investigate the expression profile of mRNA in LPS-induced bovine endometrial cells under the intervention

of IFN-τ, providing a data reference for elucidating the mechanism of action of LPS on BENDs under the influence of IFN-τ.

## 2. Materials and methods

### 2.1. Materials

Instruments: Carbon Dioxide Incubator (Shanghai Lishen Scientific Instrument Co., Ltd., HF90); Biosafety cabinet (Suzhou Antai Air Technology Co., Ltd., BSC-04IIA2); Medical refrigerator/freezer (China Science & Technology Meiling Cryogenic Technology Co., Ltd., YCD-FL450); Inverted fluorescence microscope (Ningbo Shunyu Instruments Co., Ltd., ICX41); High-speed bench centrifuge (Hunan Xiangyi Experimental Instrument Development Co., Ltd., H1750).

Reagents: Bovine endometrial epithelial cells (BENDs, BNCC359233, were purchased from Beijing Beina Chuanglian Biotechnology Research Institute, a supplier with traceable ethical procurement records. The use of these cells complies with institutional guidelines for non-animal research and does not require additional ethical approval.); recombinant sheep interferon-tau (IFN-τ, HPLC>97%, Beyotime Biotechnology, P6447); LPS (O111:B4, Sigma-Aldrich, L5293).

### 2.2. Establishment of in vitro cell models

**2.2.1. Cell culture.** BENDs were cultured in DMEM-High Glucose (DMEM-H) supplemented with 10% fetal bovine serum (FBS), 100 U·mL⁻¹ penicillin, and 100 U·mL⁻¹ streptomycin in T75 flasks at 37°C under 5% $CO_2$. The medium was replaced every 24 h, and cells were passaged every 48 h. Experiments were performed using P4-passaged cells (fourth-passaged cells).

**2.2.2. Cell grouping and processing.** BENDs were divided into a control group (C group), an LPS-treated group (L group), and an IFN-τ+LPS-treated group (F group). BENDs were cultured to the fourth generation, which were in the logarithmic growth phase and in good growth condition. After synchronization, the cells were evenly seeded in T75 cell culture flasks, and 15 mL of culture medium was added to each flask. When the cell density reaches 70%−80% in the cell culture incubator, add 40 ng/mL IFN-τ for 1 h of pre-treatment. The same volume of PBS to the control group, then stimulate the cells with 2 μg/mL LPS for 24 h. After the stimulation period, collect the cell samples.

### 2.3. Transcriptomics sequencing analysis

Three cell samples were selected from each of the control group, LPS-treated group, and IFN-τ+LPS-treated group. After cryopreservation with liquid nitrogen, the samples were sent to Wuhan Baiyi Hui Neng Biotechnology Co., Ltd. for mRNA-Seq sequencing (mRNA-Seq) analysis. The software version and parameters used for differential expression analysis were DESeq2, with a threshold of |log2FC|>1 and padj<0.05.

**2.3.1. Total RNA extraction and mRNA library construction.** Use the TRIzo1 extraction method to extract RNA from cells, collect the RNA using a centrifuge column, and dissolve the extracted RNA in Elution Buffer (Buffer No. 5). 1% agarose gel was used to monitor RNA degradation and contamination, a NanoDrop spectrophotometer was used to check RNA purity, Qubit 4.0 was used for RNA concentration and total quantity quality control, and Agilent 2100 Bioanalyzer was used to assess RNA integrity. Finally, samples that passed quality inspection were used for subsequent library construction experiments.

Before constructing an mRNA library, it is essential to verify that the RNA sample quality meets acceptable standards. Excessive rRNA can affect sequencing depth. In order to maximize the retention of mRNA containing PolyA tails, rRNA must be removed from the total RNA sample after the RNA sample has been tested and approved. First, to achieve uniform sequencing coverage, a metal ion solution treatment method was employed, and appropriate fragmentation conditions were selected based on the required fragment size for RNA fragmentation. Step 2: Using fragmented RNA as a template and random oligonucleotides as primers, synthesize the first strand of cDNA in a reverse transcriptase

system. After the first strand is synthesized, perform second-strand synthesis, end repair, and A-tail addition. Use T4 DNA ligase for adapter ligation. DNA purification beads are used to purify the adapter-ligated products, and cDNA fragments of approximately 300–350 bp are ultimately selected, ensuring the concentration of the fragments. The third step involves PCR amplification and sample labeling, followed by purification of the PCR products using DNA purification magnetic beads to obtain the library. After the library is approved, sequencing is performed.

**2.3.2. Raw data processing and quality control.** The samples were sequenced using the machine, and image files were obtained. These were converted using the sequencing platform's built-in software to generate raw data in FASTQ format, i.e., the off-machine data. The downloaded data contains some low-quality reads with junctions, which can significantly interfere with subsequent information analysis. Therefore, the sequencing data needs to be further filtered using fastp (v0.21.0). The main standards for data filtering include: remove sequences with adapters at the 3' end and remove reads with an average quality score lower than Q20. All subsequent analyses are based on high-quality clean data.

**2.3.3. Comparative analysis.** Use HISAT2 software to align the filtered reads to the reference genome. This step aims to verify the suitability of the selected reference genome and check for potential contamination or errors in the RNA extraction and sequencing processes.

**2.3.4. Overall quality assessment of the transcriptome.** Sequencing saturation analysis: Transcripts with different expression levels require varying amounts of data for effective detection. Lowly expressed genes need larger datasets to ensure detection accuracy, while highly expressed genes require less data to reach saturation during sequencing. The saturation curve illustrates the accuracy of gene expression detection under different sequencing conditions.

Gene coverage uniformity analysis: Gene coverage uniformity analysis comprehensively evaluates the sequence coverage across all genes in a sample, serving as a quality assessment for the uniformity of sequencing experiments. This analysis was performed using RSeQC-2.3.2 software.

**2.3.5. mRNA expression analysis.** Use featureCounts (v2.0.0) to count mRNA reads at the transcript level to obtain the raw expression levels, then normalize expression values using TPM (Transcripts Per Million), and finally visualize the distribution of gene/transcript expression levels across sample groups using box plots.

**2.3.6. mRNA differential analysis.** Differential expression analysis between the two comparison groups was performed using DESeq2 software (1.30.1). mRNA expression differential analysis was conducted with DESeq2, using screening criteria of $|log2FoldChange| > 1$ and $p$ adjust $\leq 0.05$ for significantly differentially expressed mRNAs.

**2.3.7. Differential gene enrichment analysis.** After completing the differential gene analysis, classify the differentially expressed genes (DEGs) between groups were classified based on genome annotation information to analyze their functional involvement.

GO enrichment analysis was conducted using the GO database, where gene lists and counts per term were calculated from DEGs annotated with GO terms. $p$-values were calculated using the hypergeometric distribution method, with a significance threshold of $p < 0.05$. Significantly enriched GO terms were identified to determine the primary biological functions associated with DEGs.

KEGG pathway enrichment analysis was conducted using clusterprofiler, where gene lists and gene numbers for each pathway were calculated from DEGs annotated to KEGG pathways. $p$-values were calculated using the hypergeometric distribution method, with significance defined at $p < 0.05$. Significantly enriched KEGG pathways were identified to reveal the signaling pathways and biological mechanisms associated with DEGs.

## 3. Results

### 3.1. RNA quality control

**3.1.1. RNA extraction quality control results.** The RIN value is an indicator of RNA integrity, ranging from 0 to 10. The higher the RIN value, the better the sample quality. The RIN values of all BEND bovine endometrial epithelial

cell samples submitted for testing in this experiment were greater than 9, indicating that the submitted samples were of acceptable quality and met the quality requirements for library construction and sequencing. The quality control results are shown in Table 1.

**3.1.2. Sequencing data quality control results.** This analysis completed whole-transcriptome sequencing for 9 samples. After filtering the sequencing data, reads with sequencing adapters and low-quality reads were removed to obtain clean reads. Bowtie was used to map the clean reads to the reference genome, yielding the genomic mapping results for each sample. The Q20 base distribution ranged from 99.07% to 99.17% (Q20 > 90% was considered acceptable), Q30 bases ranged from 97.07% to 97.35% (Q30 > 80% was considered acceptable), and GC content ranged from 47.56% to 49.83%, as shown in S1 Table.

**3.1.3. Clean reads base quality distribution results.** The x-axis of the base quality distribution plot represents the base position in the 5'-3' direction in Clean Reads, while the y-axis represents the base Q value at the corresponding site. The calculation formula is: Q = Phred = -log 10 (error rate). Q20 represents an accuracy rate of 99%, while Q30 represents an accuracy rate of 99.9%. As shown in S1 Fig in S1 File, the Q values of most sequences in the nine samples are above 30, indicating that the sequencing accuracy is at a high level.

**3.1.4. Clean reads xase content distribution results.** The x-axis of the base content distribution chart represents the base position in the 5'-3' direction in Clean Reads, while the y-axis represents the statistical proportion of a particular base at that site. Theoretically, the GC content and AT content are equal in each sequencing cycle of RNA-Seq, so the y-axis remains essentially flat throughout the sequencing process. In existing high-throughput sequencing technologies, the 15–19 bp random primers used during reverse transcription to synthesize cDNA can cause fluctuations in the nucleotide composition of the first few positions, which is considered normal. As shown in S2 Fig in S1 File, except for fluctuations in the first few positions, the y-axis of the 9 sample double-ended sequencing results remained basically horizontal, indicating that the base content was consistent with the theoretical distribution.

**3.1.5. Comparison results statistics.** The comparison results are summarized in Table 2. Using HISAT2 software to compare the clean data with the reference genome, the percentage of sequencing sequences mapped to the reference genome in the experiment was above 90% (the pass standard is > 70%). Among them, the percentage of sequencing sequences with multiple mappings was less than 10% of the total (the pass standard is usually <10%), indicating that the reference genome was appropriately selected and there was no contamination in the operation.

**3.1.6. Sequencing saturation analysis.** The x-axis of the saturation analysis graph represents the ratio of resampled mapped reads, while the y-axis represents the relative error between gene expression levels at that ratio and actual expression levels. Q1, Q2, Q3, and Q4 represent the distribution of gene expression levels (RPKM) in the 0–25% percentile, 25–50% percentile, 50–75% percentile, and 75–100% percentile, respectively. As shown in S3 Fig in

**Table 1. RNA extraction quality control results.**

| Sample name | Concentration (ng/µL) | Volume (µL) | Total amount (ug) | RIN | Result |
|---|---|---|---|---|---|
| C1 | 1254 | 22 | 27.588 | 9.30 | Pass |
| C2 | 1189 | 22 | 26.158 | 9.20 | Pass |
| C3 | 1481 | 22 | 32.582 | 9.20 | Pass |
| L1 | 960 | 22 | 21.120 | 9.20 | Pass |
| L2 | 1268 | 22 | 27.896 | 9.00 | Pass |
| L3 | 1316 | 22 | 28.952 | 9.10 | Pass |
| F1 | 1227 | 22 | 26.994 | 9.20 | Pass |
| F2 | 1260 | 22 | 27.720 | 9.00 | Pass |
| F3 | 1200 | 22 | 26.400 | 9.00 | Pass |

**Table 2. Comparison results statistics.**

| Sample name | Total reads | Total mapped | Multiple mapped | Uniquely mapped |
|---|---|---|---|---|
| C1 | 67229506 | 62613030 (93.13%) | 3435891 (5.11%) | 59177139 (88.02%) |
| C2 | 70432934 | 66332511 (94.18%) | 3693341 (5.24%) | 62639170 (88.93%) |
| C3 | 67727080 | 63056478 (93.10%) | 3357865 (4.96%) | 59698613 (88.15%) |
| L1 | 73516268 | 66760899 (90.81%) | 4248334 (5.78%) | 62512565 (85.03%) |
| L2 | 69812388 | 64290331 (92.09%) | 3676545 (5.27%) | 60613786 (86.82%) |
| L3 | 66674220 | 62599343 (93.89%) | 2942704 (4.41%) | 59656639 (89.47%) |
| F1 | 100743782 | 93933453 (93.24%) | 4912715 (4.88%) | 89020738 (88.36%) |
| F2 | 69687572 | 64396104 (92.41%) | 2744470 (3.94%) | 61651634 (88.47%) |
| F3 | 69084032 | 64393640 (93.21%) | 2889953 (4.18%) | 61503687 (89.03%) |

S1 File, the Percent Relative Error values under different conditions decrease as the number of resampled mapped reads increases. When the percentage is close to 100%, the Percent Relative Error values are all less than 15, indicating that the gene expression levels remain essentially unchanged and the curve has reached saturation.

**3.1.7. Gene coverage uniformity analysis.** The x-axis of the gene coverage uniformity map represents the percentage of a single gene's base length relative to the total base length, with values ranging from 0 to 100 indicating the gene's relative position in the 5'-3' direction. The y-axis represents the total number of sequences aligned to the corresponding interval on the x-axis. The curve reflects whether the sequenced sequences are uniformly distributed across the gene. If there is a noticeable peak near the right end, it indicates that the sequencing results have a noticeable 3' bias. If there is a noticeable peak near the left end, it indicates that the sequencing results have a noticeable 5' bias. As shown in S4 Fig in S1 File, the sequenced sequences are evenly distributed in the 5' to 3' region, with no obvious peaks at either end of the curve, indicating that the sequencing results are unbiased and relatively uniform.

## 3.2. Differential expression

**3.2.1. mRNA expression analysis.** Fig 1A shows the mRNA box plot. After normalization of the three groups of samples, the data distribution is average and highly reliable. mRNA expression density plots generally exhibit a non-standard normal distribution, with the area of the region equal to 1, representing a total probability of 1. The peak of the density distribution curve represents the region where the expression of the gene is most concentrated in the entire sample (Fig 1B). mRNA correlation analysis (Fig 1C) showed that the correlation coefficients between groups ranged from 0.9 to 1, indicating a high degree of similarity in expression patterns between samples.

**3.2.2. Analysis of differentially expressed mRNA genes.** Volcano plots represent mRNA expression levels, and the screening criteria for differential mRNA expression are $|Log2FC| > 1$ and $p < 0.05$ (Fig 2A). Compared with the control group, after adding LPS, 100 mRNAs were significantly upregulated and 9 mRNAs were downregulated considerably; after IFN-τ intervention, 755 mRNAs were significantly upregulated and 354 mRNAs were significantly downregulated. Compared with the LPS group, after IFN-τ intervention, 572 mRNAs were significantly upregulated and 390 mRNAs were significantly downregulated. Clustering heat map analysis of differentially expressed genes (Fig 2B) revealed genes with highly correlated expression levels between samples, some of which may be involved in biological processes, metabolic

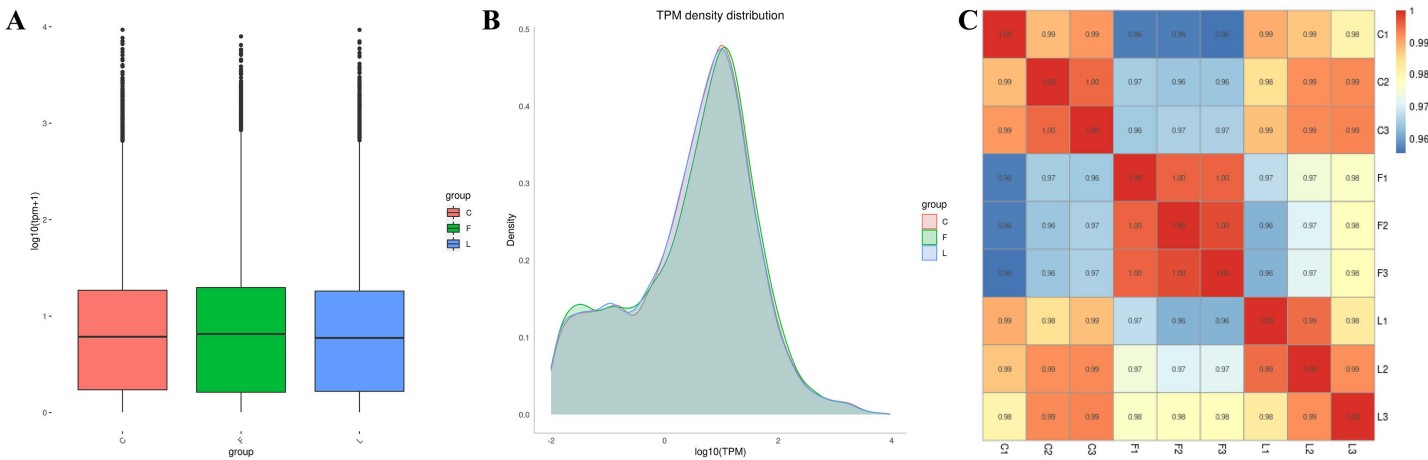

**Fig 1. (A) mRNA sample distribution map. (B)** mRNA expression density map. **(C)** correlation clustering heat map.

processes, or signaling pathways. As shown in Fig 2C, there are 61 differentially expressed genes common to all three groups. The numbers of differentially expressed genes specific to the C vs L group, C vs F group, and L vs F group are 21, 341, and 209, respectively.

### 3.3. Enrichment analysis

To explore the biological functions of differentially expressed genes, we performed GO enrichment analysis to study the molecular functions, cellular components, and biological processes most likely to be associated with differentially expressed genes. Fig 3A shows the GO enrichment bar chart of differentially expressed genes in the top 20 categories. The x-axis represents the GO terms in the next level of the three major GO categories, and the y-axis represents the number of differentially expressed genes enriched under that term (including its sub-terms). C vs L group, there are 7 GO terms annotated to biological processes (BP), 1 GO term annotated to cellular components (CC), and 12 GO terms annotated to molecular functions (MF). Mainly significantly enriched in defense response to virus, double-stranded RNA binding, extracellular region, chemokine activity, nucleotidyltransferase activity, chemotaxis, defense response, immune response, transcription cis-regulatory region binding. C vs F group, there were 11 GO terms annotated to biological processes (BP), 2 GO terms annotated to cellular components (CC), and 7 GO terms annotated to molecular functions (MF). The main significant enrichments are in antigen processing and presentation, immune response, defense response to virus, MHC class II protein complex, lipoprotein metabolic process, double-stranded RNA binding, complement activation, lipid transport, NAD+ADP-ribosyltransferase activity, and autophagy. L vs F group, there were 13 GO terms annotated to biological processes (BP), 1 GO term annotated to cellular components (CC), and 6 GO terms annotated to molecular functions (MF). It is primarily enriched in defense responses to viruses, lipoprotein metabolic processes, double-stranded RNA binding, lipid transport, NAD+ADP-ribosyltransferase activity, GTP binding, and extracellular regions.

To investigate the KEGG pathways of differentially expressed genes, we performed KEGG enrichment analysis. As shown in Fig 3B, the top 20 differentially expressed genes are shown in the KEGG enrichment bubble chart. The x-axis in the figure represents GeneRatio, which indicates the proportion of differentially expressed genes enriched in that entry relative to the total number of differentially expressed genes in the functional annotation results. The y-axis represents the entries in the enrichment. The size of the dots represents the number of genes in the enrichment, and the color represents the $p$-value, with lower values indicating greater significance. Among these, compared with the control group, genes with significant differences in expression after LPS addition were significantly enriched in the NOD-like receptor signaling

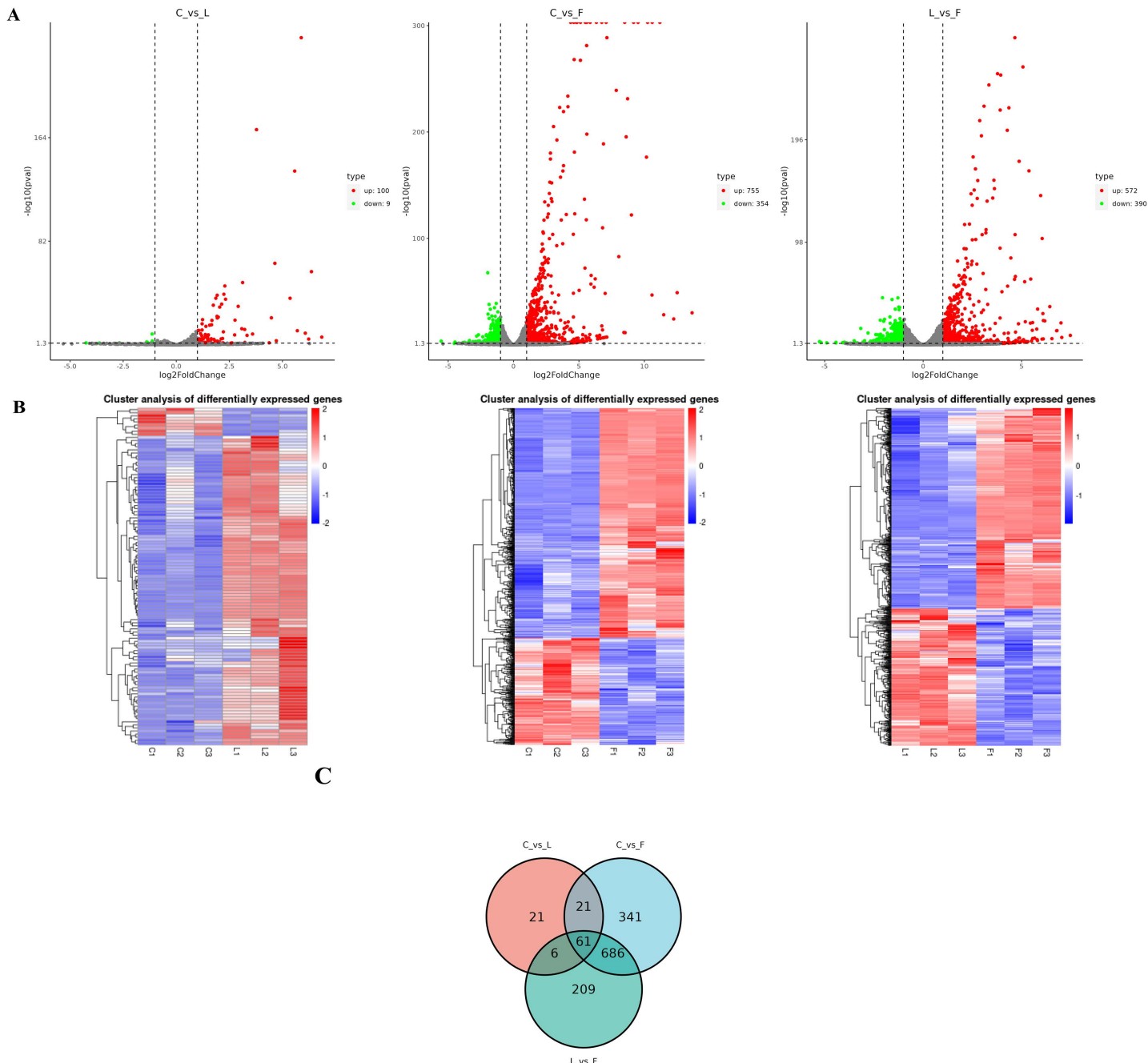

**Fig 2. Screening and analysis of differentially expressed mRNA genes. (A)** Volcano plot showing expression levels of differentially expressed genes between group C and group L (left), group C and group F (middle), and group L and group F (right).Red dots indicate differentially expressed genes that are upregulated, green dots indicate differentially expressed genes that are downregulated, and gray dots indicate genes that were detected but did not show significant differences. **(B)** Cluster heat map. Normalize the differences in genes between Group C and Group L (left), Group C and Group F (middle), and Group L and Group F (right). The horizontal axis represents group information, and the vertical axis represents differentially expressed gene information. The clustering tree on the left side of the figure is the differentially expressed gene clustering tree. Scale represents the expression level after standardization (the redder the color, the higher the expression level). **(C)** The Venn diagram illustrates the common and unique differences between genes in Group C and Group L, Group C and Group F, and Group L and Group **F.**

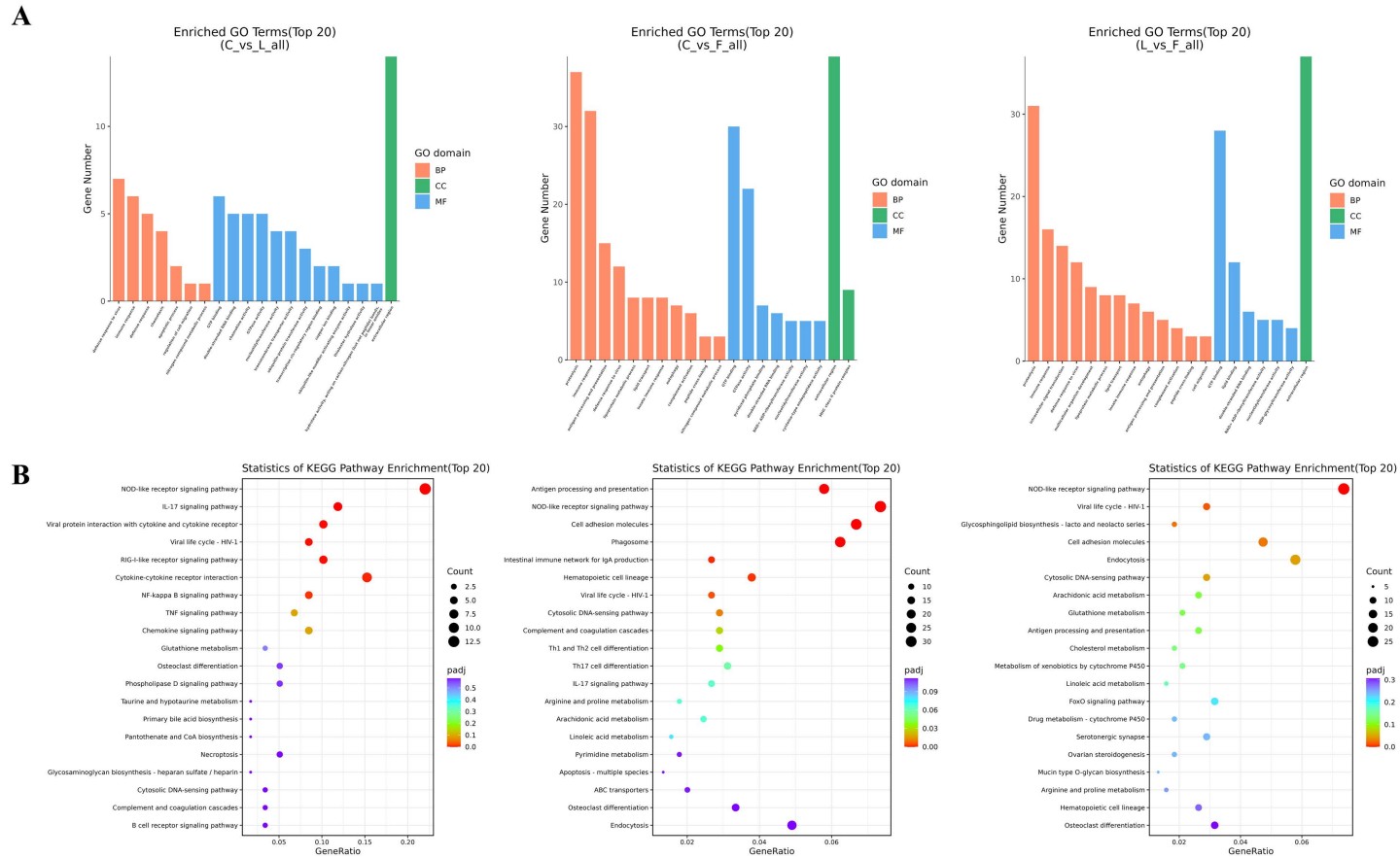

**Fig 3. Enrichment analysis of differentially expressed mRNAs. (A)** GO enrichment column chart showing the expression levels of differentially expressed genes between Group C and Group L (left), Group C and Group F (middle), and Group L and Group F (right). **(B)** KEGG enrichment bubble chart showing the expression levels of differentially expressed genes between Group C and Group L (left), Group C and Group F (middle), and Group L and Group F (right).

pathway, IL-17 signaling pathway, Viral protein interaction with cytokine and cytokine receptor, Viral life cycle-HIV-1, RIG-I-like receptor signaling pathway, Cytokine-cytokine receptor interaction, and NF-κB signaling pathway. Following IFN-τ intervention, KEGG analysis revealed that Antigen processing and presentation, NOD-like receptor signaling pathway, Cell adhesion molecules (CAM), Phagosome, Intestinal immune network for IgA production, Hematopoietic cell lineage, Viral life cycle-HIV-1, Cytosolic DNA-sensing pathway, Complement and coagulation cascades, and Th1 and Th2 cell differentiation were significantly enriched ($p < 0.05$). The significance ranking of the NF-κB signaling pathway was within 40, but there was no statistical difference ($p = 0.37433$). The significance ranking of the IL-17 signaling pathway was within 20, but there was no statistical difference ($p = 0.06457$). Compared with the LPS group, after IFN-τ intervention, the differentially expressed genes were significantly enriched in the NOD-like receptor signaling pathway, viral life cycle-HIV-1, glycosphingolipid biosynthesis-lacto and neolacto series, Cell adhesion molecules (CAM), the Cytosolic DNA-sensing pathway, and Endocytosis. The significance ranking of the NF-κB signaling pathway was within 100, but there was no statistical difference ($p = 0.85268$). The significance ranking of the IL-17 signaling pathway was within 105, but there was no statistical difference ($p = 0.86409$).

## 4. Discussion

Currently, RNA-seq sequencing technology has been widely used in the study of the occurrence and development of animal immunity and inflammation. By utilizing high-throughput sequencing technology for cDNA sequencing, it is possible to comprehensively and rapidly obtain information on nearly all transcripts in a specific species' tissue or cells under a given state. Additionally, bioinformatics techniques are employed to analyze gene expression differences and gene structural variations, thereby providing precise data information for life science research. Transcriptome sequencing, functional gene research, and differential gene analysis of bovine endometritis are of great practical significance for studying the pathogenesis of bovine endometritis and can also provide new ideas and methods for the prevention and treatment of bovine endometritis.

This study utilized RNA-seq sequencing technology to analyze the transcriptomic information of the control group (C group), LPS group (L group), and IFN-τ+LPS group (F group) endometrial cells from dairy cows. Compared with the control group, the addition of LPS resulted in 109 differentially expressed genes, including 100 upregulated and 9 downregulated genes. Following IFN-τ intervention, a total of 1,109 differentially expressed genes were identified, comprising 755 upregulated genes and 354 downregulated genes. Compared with the LPS group, a total of 962 differentially expressed genes were identified after IFN-τ intervention, with 572 genes upregulated and 390 downregulated. Differentially expressed genes were subjected to KEGG pathway enrichment analysis, which identified significantly enriched signaling pathways, including those related to immunity, such as the NOD-like receptor signaling pathway, cytokine-cytokine receptor interactions, NF-κB signaling pathway, IL-17 signaling pathway, and RIG-I-like receptor signaling pathway. Additionally, the FoxO signaling pathway, chemokine signaling pathway, B-cell receptor signaling pathway, p53 signaling pathway, MAPK signaling pathway, PI3K-Akt signaling pathway, Jak-STAT signaling pathway, and other immune-related signaling pathways also showed enrichment of differentially expressed genes.

Pattern recognition receptors (PRRs) are a key component of the host innate immune system. They recognize specific pathogen-associated molecular patterns (PAMPs) to initiate intracellular signaling cascades that induce the secretion of inflammatory cytokines and chemokines, ultimately eliminating invading pathogens and infected cells [24]. Following recognition by receptors on the membrane surface, LPS triggers an inflammatory response within cells, activating intracellular signaling cascades. The identified differentially expressed genes (DEGs) are primarily concentrated in the following pathways: cytokine-cytokine receptor interactions, IL-17, TNF, NOD-like receptors, RIG-I-like receptors, chemokines, TLR, NF-κB signaling, and the cytoplasmic DNA sensing pathway [25–28]. It has been reported that NF-κB is closely associated with calcium signaling pathways, Toll-like receptors, T cell/B cell receptors, and cell-cytokine receptor interactions, and can be activated by MHC-presented antigens, bacterial and viral antigens, LPS, and cytokines [29]. The main member of the IL-17 family, IL-17A, is secreted by Th17 cells and CD8+cells [30,31] and induces the production of various proinflammatory factors, including cytokines and chemokines, through cell-cytokine receptors, the NF-κB, and MAPK cascades [32], playing a crucial role in inflammatory responses. These pathways lead to the expression of inflammatory mediators and contribute to the clearance of pathogens [24]. CD38, as a NAD+ hydrolase, directly reduces intracellular NAD+ levels by catalyzing the conversion of NAD+ into second messengers such as cADPR and NAADP. Research by Da Liu et al. revealed that following noise exposure, the levels of nicotinamide adenine dinucleotide (NAD) in metabolism decreased, accompanied by activation of the NF-κB signaling pathway. Expression of CD38, the primary NAD hydrolase in mammals, may directly contribute to inflammation. Apigenin significantly reduced noise-induced hearing loss thresholds by inhibiting NF-κB and CD38 expression, suggesting CD38 may serve as an effective therapeutic target for noise-induced hearing loss [33]. HADHA, as a key enzyme in fatty acid β-oxidation, is essential for maintaining cellular energy balance [34]. Changes in HADHA expression and activity affect cellular energy supply and lipid metabolite accumulation, thereby triggering apoptotic signaling pathways and subsequently influencing the functional state of endometrial cells in an inflammatory context [35–37]. Research by Guoshang Ji et al. revealed that HADHA suppresses proliferation of bovine endometrial epithelial cells by regulating key signaling

pathways including TGF-β signaling, fatty acid metabolism, p53 signaling, and IL-17 signaling, providing insights into the molecular mechanisms underlying bovine endometritis [38].

Additionally, our data further corroborate these findings, demonstrating that LPS triggers cytokine-cytokine receptor interactions and signaling pathways involving NOD-like receptors, IL-17, RIG-I-like receptors, NF-κB, and others, leading to the onset of inflammation in the body. In this study, we found that IFN-τ intervention altered mRNA expression in bovine endometrial epithelial cells, with differentially expressed genes primarily concentrated in Antigen processing and presentation, Cell adhesion molecules (CAM), glycosphingolipid biosynthesis-lacto and neolacto series, Phagosome, Complement and coagulation cascades, and Endocytosis. Similar to previous reports, the complement signaling cascade is also responsible for immune defense, with complement activation leading to pathogen phagocytosis and clearance via phagocytes and cell lysis [39]. Cell adhesion molecules (CAMs) mediate cell-cell and cell-extracellular matrix (ECM) interactions and binding, and the expression of their associated genes and proteins plays a crucial role in regulating immune responses and tissue repair [40,41]. Glycosphingolipids (GSLs) are a group of membrane components primarily found on the surface of mammalian cells. They and their metabolites play a role in intercellular communication, acting as multifunctional biochemical signals and in many cellular pathways, and play a key role in pain and inflammatory responses [42,43]. T cells recognize exogenous antigens through T cell antigen receptors (TCRs) and play a central role in acquired immunity. Following antigen stimulation, CD4 + T cells rapidly proliferate and differentiate into functionally distinct subsets, including Th1, Th2, Th17, and regulatory T cells (Tregs) [44,45]. The sphingolipid biosynthetic pathway is typically activated in Th1, Th2, Th17, and iTreg cells, and inhibition of this biosynthetic pathway leads to suppression of Th17 and iTreg cell differentiation [46]. Alam S [47] et al. showed that the new lactose series of sphingolipids (nsGSL) is an important group of sphingolipids (GSL) that may be involved in specific cellular physiological functions.

## 5. Conclusion

This study investigated how IFN-τ regulates microbial-induced endometritis using an LPS-treated bovine endometrial cell model. RNA-seq analysis showed LPS activates key inflammatory pathways (NOD-like receptor, NF-κB, IL-17) within 24 hours, confirming cellular responses to infection. IFN-τ intervention suppressed these pathways by enhancing pathogen recognition through cell adhesion molecules and optimizing immune clearance processes (antigen presentation, complement cascades, phagocytosis). It also promoted pathogen clearance by modulating glycosphingolipid biosynthesis. These findings reveal IFN-τ's multi-pathway network for anti-inflammatory effects, offering theoretical support for its clinical use in treating endometritis and improving reproductive outcomes.

## Supporting information

**S1 File. S1 Fig. Clean Reads base quality distribution.** S2 Fig. Distribution of base content in clean reads. S3 Fig. Saturation analysis. S4 Fig. Gene coverage uniformity.
(ZIP)

**S1 Table. Statistics on sequencing data quality control results.**
(DOCX)

## Acknowledgments

The authors would like to thank Tarim University, China, for providing research equipment and experimental materials. The authors acknowledge the assistance of the Key Laboratory of Livestock and Forage Resources Utilization in the Tarim Basin, Ministry of Agriculture and Rural Affairs.

## Author contributions

**Conceptualization:** Pan Liu, Yaofeng Zhang, Xu Chen, Lijun Tang, Junfeng Liu.

**Data curation:** Pan Liu, Yaofeng Zhang, Xu Chen, Lijun Tang.

**Formal analysis:** Pan Liu.

**Funding acquisition:** Junfeng Liu.

**Investigation:** Wenxiu Xu, Guigui Wang, Duoqi Zhang.

**Methodology:** Wenxiu Xu, Guigui Wang.

**Project administration:** Junfeng Liu.

**Supervision:** Junfeng Liu.

**Validation:** Duoqi Zhang, Junfeng Liu.

**Writing – original draft:** Pan Liu, Yaofeng Zhang.

**Writing – review & editing:** Junfeng Liu.

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
