## [Decision Letter · Decision Letter 0]

23 Dec 2025

Exploring the effects of IFN-τ on LPS-induced endometritis in cows based on transcriptomics

PLOS One

Dear Dr. Liu,

Thank you for submitting your manuscript to PLOS ONE. After careful consideration, we feel that it has merit but does not fully meet PLOS ONE’s publication criteria as it currently stands. Therefore, we invite you to submit a revised version of the manuscript that addresses the points raised during the review process.

We look forward to receiving your revised manuscript.

Kind regards,

Vikash Chandra, PhD

Academic Editor

PLOS One

Journal Requirements:

“National Natural Science Foundation of China [grant numbers 32260905]”

Please state what role the funders took in the study.  If the funders had no role, please state: 'The funders had no role in study design, data collection and analysis, decision to publish, or preparation of the manuscript.'

4. Please note that funding information should not appear in any section or other areas of your manuscript. We will only publish funding information present in the Funding Statement section of the online submission form. Please remove any funding-related text from the manuscript.

5. Thank you for uploading your study's underlying data set. Unfortunately, the repository you have noted in your Data Availability statement does not qualify as an acceptable data repository according to PLOS's standards.

8 Please review your reference list to ensure that it is complete and correct. If you have cited papers that have been retracted, please include the rationale for doing so in the manuscript text, or remove these references and replace them with relevant current references. Any changes to the reference list should be mentioned in the rebuttal letter that accompanies your revised manuscript. If you need to cite a retracted article, indicate the article’s retracted status in the References list and also include a citation and full reference for the retraction notice.

Additional Editor Comments:

The manuscript "Exploring the effects of IFN-τ on LPS-induced endometritis in cows based on transcriptomics" is an interesting investigation by the authors, conducted systematically with valuable findings. However, there are some points to be addressed before acceptance for publication as also pointed by the reviewers. So, it is recommended for a minor revision. In addition to the reviewers' comments, address the following points: 1. Correct the short title, 2. Insert literature about the clinical application of IFN-tau in endometritis (if available or justify your point), 3. Reduce the number of references, limiting it to 40 (delete less relevant citations), 4. Improve English grammar for better readability.

Reviewer's Responses to Questions

**Comments to the Author**

1. Is the manuscript technically sound, and do the data support the conclusions?

Reviewer #1: Yes

Reviewer #2: Yes

2. Has the statistical analysis been performed appropriately and rigorously?

Reviewer #1: Yes

Reviewer #2: Yes

3. Have the authors made all data underlying the findings in their manuscript fully available?

Reviewer #1: Yes

Reviewer #2: Yes

4. Is the manuscript presented in an intelligible fashion and written in standard English?

Reviewer #1: Yes

Reviewer #2: Yes

Reviewer #1: Manuscript ID: PONE-D-25-59265

Title: Exploring the effects of IFN-τ on LPS-induced endometritis in cows based on transcriptomics

The manuscript entitled “Exploring the effects of IFN-τ on LPS-induced endometritis in cows based on transcriptomics” describes the transcriptomic profile in bovine endometrial cell culture simulating LPS induced endometritis. Further, the effect of IFN-� on the modulation of transcript profile in LPS induced endometritis in bovine endometrial epithelial cell culture model. The manuscript is well designed and describes the experimental methodology and illustrated the significant findings in terms of differentially expressed genes and their associated biological pathway.

However, the rationale behind the use of IFN� as possible therapeutic target in endometritis is not clear. It is a proven fact that IFN� is a key signal molecule for maternal recognition of pregnancy in ruminants. It is produced by the mononuclear trophoblast cells during peri implantation period and suppress PGF2� release from the endometrium by modulating expression of oxytocin receptor (OTR) and Estradiol receptor. Thus, IFN� prevent luteolysis to maintain progesterone secretion from the corpus luteum required for establishment and maintenance of pregnancy. Progesterone has immunosuppressive effect on endometrium and aggravates the endometritis further. on the other hand, in bovine endometritis, PGF2� is considered as a drug of choice as it causes luteolysis and induces estrus, thus helps in reversal of progesterone effect.

In present study, the role of IFN� as a possible therapeutic option in bovine endometritis emphasizing only the anti-inflammatory role overlooking its role in modulating key steroidogenic pathway and their receptor expression limits its relevance as therapeutics in clinical use. It would be better to relook the transcriptomic profile considering the biological relevance of IFN-� in LPS induced bovine endometritis model and to explore the risk of MRP and increase chances of embryonic mortality in endometritis cows.

The rationale of the study needs to be focused considering these facts and the manuscript need to be revised thoroughly. Following points are to be considered for the revision.

1. Many abbreviations were used throughout the text without mentioning their expanded form. Always use the expanded form of the abbreviations when used for the first time.

2. Line 40: microorganisms invading the uterus through the open cervix during pregnancy. Check this statement. During pregnancy cervix remains closed and it prevents entry of pathogens into the uterus. While pathogen may get entry into the uterus through open cervix at estrus or during parturition.

3. The rationale behind use of IFN-� and LPS in the transcriptomic profiling in bovine endometrial epithelial cell culture model is not clear.

It is well established fact that in endometritis animal chances of embryonic survival is low due to inflammatory reactions in the endometrium, production of inflammatory mediators including PGF2� and failure of MRP due to sub-optimal production of IFN�. As IFN� is key signaling molecule for MRP, It would be better to highlight the importance of IFN� induced transcriptomic profile in LPS treated bovine endometrial epithelial cell culture model to understand the pathophysiology of early embryonic mortality in bovine endometritis.

4. In the experimental design three groups, Control (C), LPS (L) and IFN� + LPS (F) groups were included. To understand the effect of IFN� on the transcriptomic profile an additional group of IFN� alone is required.

5. Line #198-202: The Q20 base distribution ranged from 98.95% to 99.57% (Q20 > 90% was considered acceptable), Q30 bases ranged from 96.45% to 97.98% (Q30 > 80% was considered acceptable), and GC content ranged from 41.99% to 56.48%.

However, the values mentioned in Suppl. Table 1. does not match with the values mentioned in the text. Please check and clarify.

6. In the introduction and discussion, include relevant literature supporting the role of IFN � in modulating transcriptomic profile in MRP and during embryonic mortality in endometritis cows.

7. Conclusion: Concise the conclusion focusing the most significant findings and possible clinical applications in bovine endometritis.

Reviewer #2: The manuscript presents a well-designed transcriptomic study investigating the anti-inflammatory role of IFN-τ in LPS-induced bovine endometritis. The research question is clear, the methodology is robust, and the findings are supported by appropriate bioinformatic analyses. The study contributes meaningfully to understanding the molecular mechanisms of endometritis and potential therapeutic applications of IFN-τ. However, several editorial and formatting issues must be addressed to meet PLOS ONE.

Major Concerns

1. Figures and tables are referenced in the text (e.g., Fig. 1, Table 1), but the actual figures are not embedded in the submitted PDF (only file names are listed). Ensure all figures and tables are included in the manuscript file or as separate high-resolution files.

2. The statement is appropriate and includes an accession number (PRJNA1337137). Ensure the link is functional and that data will be publicly available upon publication.

3. The manuscript uses a commercial bovine endometrial epithelial cell line (BENDs). While this does not require IACUC approval, a clear statement confirming the cell line’s ethical sourcing and absence of animal experimentation should be provided in the Methods and the Ethics Statement section (currently “N/A”).

Minor Concerns

Short Title: Incorrectly lists section headers. Replace with a short, descriptive title.

Keywords: Should be listed as: endometritis; IFN-τ; transcriptomics; LPS (currently included in abstract).

Abbreviations: Define all abbreviations at first use (e.g., BENDs, LPS, IFN-τ, DEmRNAs, GO, KEGG).

Inconsistent Symbols: Use “IFN-τ” consistently (not “IFN-r”).

Page 10, Line 10: “Glycosphingolipid biosynthesis-lacto and neolacto series” – ensure spelling consistency (“lacto” vs. “lacta”).

Cell Culture: Specify the source and passage number of BEND cells.

RNA-seq Analysis: Mention software versions and parameters used for differential expression analysis (e.g., DESeq2, threshold: |log2FC| > 1, padj ≤ 0.05).

Ethical Compliance: Add a sentence confirming that no live animals were used and that the cell line was obtained commercially.

Clarity: Some descriptions are dense. Consider breaking into subsections (e.g., 3.1. RNA Quality Control, 3.2. Differential Expression, 3.3. Enrichment Analysis).

Supplementary Material: Ensure all supplementary figures/tables are cited in the text (e.g., Supplementary Fig. 1, Supplementary Table 1).

The discussion is comprehensive but could be more concise. Focus on integrating findings with existing literature and highlighting novel insights from this study.

Format references according to PLOS ONE style (e.g., Vancouver style). Ensure all in-text citations match the reference list.

**Do you want your identity to be public for this peer review?** For information about this choice, including consent withdrawal, please see our Privacy Policy

Reviewer #1: **Yes:** Manas Kumar Patra

Reviewer #2: No

---

## [Author Response · Author response to Decision Letter 1]

20 Jan 2026

Dear Editors and Reviewers:

Thank you for your letter and for the reviewers, comments concerning our manuscript entitled “Exploring the effects of IFN-τ on LPS-induced endometritis in cows based on transcriptomics” (Manuscript Number: PONE-D-25-59265 ). Those comments are all valuable and very helpful for revising and improving our paper, as well as the important guiding significance to our researches. We have studied comments carefully and have made correction which we hope meet with approval. Revised portion are marked in red in the manuscript. The main corrections in the manuscript and the responds to the reviewer’s comments are as flowing:

Journal Requirements:

1.Please ensure that your manuscript meets PLOS ONE's style requirements, including those for file naming.

Response: Thank you for reviewing the manuscript and providing valuable feedback. We have revised it according to the journal's formatting requirements.

2.Please note that PLOS One has specific guidelines on code sharing for submissions in which author-generated code underpins the findings in the manuscript. In these cases, we expect all author-generated code to be made available without restrictions upon publication of the work. Please review our guidelines at https://journals.plos.org/plosone/s/materials-and-software-sharing#loc-sharing-code and ensure that your code is shared in a way that follows best practice and facilitates reproducibility and reuse.

Response: Thank you for your constructive feedback. The “Transcriptomics Sequencing Data - mRNA-Seq sequencing data” in this study are available in the National Center for Biotechnology Information (NCBI). The project number is PRJNA1337137 (https://www.ncbi.nlm.nih.gov/sra/PRJNA1337137). And the data is now publicly available.

3. Thank you for stating the following financial disclosure: “National Natural Science Foundation of China [grant numbers 32260905]” Please state what role the funders took in the study.  If the funders had no role, please state: 'The funders had no role in study design, data collection and analysis, decision to publish, or preparation of the manuscript.' If this statement is not correct you must amend it as needed. Please include this amended Role of Funder statement in your cover letter; we will change the online submission form on your behalf.

Response: Thank you for your guidance regarding the Role of Funder statement. We appreciate your attention to this important aspect of our manuscript submission. As requested, we have revised the Role of Funder statement to clarify that: "The National Natural Science Foundation of China [grant number 32260905] had no role in study design, data collection and analysis, decision to publish, or preparation of the manuscript." This revised statement has been included in our cover letter for this submission. We kindly request your assistance in updating the online submission form accordingly. Please let us know if any further adjustments are needed. We value your support in ensuring compliance with journal requirements.

4.Please note that funding information should not appear in any section or other areas of your manuscript. We will only publish funding information present in the Funding Statement section of the online submission form. Please remove any funding-related text from the manuscript.

Response: We have removed any references to funding from the manuscript.

5.Thank you for uploading your study's underlying data set. Unfortunately, the repository you have noted in your Data Availability statement does not qualify as an acceptable data repository according to PLOS's standards. At this time, please upload the minimal data set necessary to replicate your study's findings to a stable, public repository (such as figshare or Dryad) and provide us with the relevant URLs, DOIs, or accession numbers that may be used to access these data. For a list of recommended repositories and additional information on PLOS standards for data deposition, please see https://journals.plos.org/plosone/s/recommended-repositories.

Response: Thank you for your constructive feedback. The “Transcriptomics Sequencing Data - mRNA-Seq sequencing data” in this study are available in the National Center for Biotechnology Information (NCBI). The project number is PRJNA1337137 (https://www.ncbi.nlm.nih.gov/sra/PRJNA1337137). And the data is now publicly available.

We sincerely appreciate your vigilance on data transparency. Please let us know if you require additional details regarding the data or if further adjustments are needed.

6.When completing the data availability statement of the submission form, you indicated that you will make your data available on acceptance. We strongly recommend all authors decide on a data sharing plan before acceptance, as the process can be lengthy and hold up publication timelines. Please note that, though access restrictions are acceptable now, your entire data will need to be made freely accessible if your manuscript is accepted for publication. This policy applies to all data except where public deposition would breach compliance with the protocol approved by your research ethics board. If you are unable to adhere to our open data policy, please kindly revise your statement to explain your reasoning and we will seek the editor's input on an exemption. Please be assured that, once you have provided your new statement, the assessment of your exemption will not hold up the peer review process.

Response: We fully understand and support PLOS ONE's open data policy. Regarding the situation you mentioned—declaring in the submission form that data will be provided upon acceptance—we have already completed data sharing in advance. The “Transcriptomics Sequencing Data - mRNA-Seq sequencing data” in this study are available in the National Center for Biotechnology Information (NCBI). The project number is PRJNA1337137 (https://www.ncbi.nlm.nih.gov/sra/PRJNA1337137).

7.If the reviewer comments include a recommendation to cite specific previously published works, please review and evaluate these publications to determine whether they are relevant and should be cited. There is no requirement to cite these works unless the editor has indicated otherwise.

Response: We appreciate the editor's valuable suggestions. Should the review comments include recommendations to cite specific published literature, we will review and evaluate these references to determine their relevance and whether they should be included.

8.Please review your reference list to ensure that it is complete and correct. If you have cited papers that have been retracted, please include the rationale for doing so in the manuscript text, or remove these references and replace them with relevant current references. Any changes to the reference list should be mentioned in the rebuttal letter that accompanies your revised manuscript. If you need to cite a retracted article, indicate the article’s retracted status in the References list and also include a citation and full reference for the retraction notice.

Response: We appreciate the editor's valuable suggestions and guidance regarding literature citations. We have verified the reference list to ensure no retracted publications are cited. Any modifications to the reference list are fully documented in the response letter accompanying the revised manuscript.

Additional Editor Comments:

1.Correct the short title

Response: Thank you for your attention to detail regarding the manuscript. Regarding your request to “Correct the short title,” we have immediately verified and completed the modification.

2.Insert literature about the clinical application of IFN-tau in endometritis (if available or justify your point)

Response: Thank you for your meticulous review of the manuscript content and attention to research details. Regarding your suggestion to “Insert literature about the clinical application of IFN-tau in endometritis (if available or justify your point),” we through systematic searches of databases including PubMed, Web of Science, and ClinicalTrials.gov, we found that: Current research on IFN-tau (interferon tau) primarily focuses on animal models and cell models. No publicly published clinical trials or application cases involving IFN-tau have been identified. Existing studies predominantly explore its potential as a biomarker or therapeutic target rather than direct clinical intervention.

3.Reduce the number of references, limiting it to 40 (delete less relevant citations)

Response: Thank you for your meticulous review of the references section in our manuscript. Regarding your suggestion to “Reduce the number of references, limiting it to 40 (delete less relevant citations),” we have made the necessary deletions. After these revisions, the total number of references has been reduced from 53 to 47, though it still exceeds your recommended limit of 40. According to PLOS ONE's citation policy, the number of references should be “sufficient to support the research content,” rather than mechanically adhering to numerical limits. The current 47 citations represent a streamlined result, and further reduction may compromise the integrity of the argumentation.

4.Improve English grammar for better readability.

Response: Thank you for your attention to the linguistic quality of the manuscript. We have commissioned a native English speaker to perform systematic revisions of English grammar, enhancing readability. Should you believe certain paragraphs require further refinement, we will implement targeted adjustments based on your specific feedback.

Reviewer #1:

1.Many abbreviations were used throughout the text without mentioning their expanded form. Always use the expanded form of the abbreviations when used for the first time.

Response: Thank you for pointing this out! As the author, I will strictly adhere to academic writing standards and ensure that the full form of any abbreviation is provided upon its first use in the text. The following modifications are now made: Line 16 Lipopolysaccharide (LPS); Line 18 Interferon-tau (IFN-τ); Line 21 bovine endometrial cells (BENDs); Line 24-25 Differentially Expressed mRNAs (DEmRNAs); Line 25 Gene Ontology (GO); Line 26 Kyoto Encyclopedia of Genes and Genomes (KEGG).

2.Line 40: microorganisms invading the uterus through the open cervix during pregnancy. Check this statement. During pregnancy cervix remains closed and it prevents entry of pathogens into the uterus. While pathogen may get entry into the uterus through open cervix at estrus or during parturition.

Response: Thank you for your meticulous review and insightful feedback. We sincerely apologize for the conceptual error in our original statement regarding cervical status during pregnancy, which we acknowledge as a oversight in our manuscript.

Correction Made:

Original text (Line 40):

Endometritis is a reproductive system disease in dairy cows caused by microorganisms invading the uterus through the open cervix during pregnancy or after calving.

Revised text:

Endometritis is a reproductive system disease in dairy cows caused by microorganisms invading the uterus through the open cervix during estrus or after calving. (Line 42-44)

3. The rationale behind use of IFN-τ� and LPS in the transcriptomic profiling in bovine endometrial epithelial cell culture model is not clear. It is well established fact that in endometritis animal chances of embryonic survival is low due to inflammatory reactions in the endometrium, production of inflammatory mediators including PGF2� and failure of MRP due to sub-optimal production of IFN-τ�. As IFN-τ� is key signaling molecule for MRP, It would be better to highlight the importance of IFN-τ induced transcriptomic profile in LPS treated bovine endometrial epithelial cell culture model to understand the pathophysiology of early embryonic mortality in bovine endometritis.

Response: Thank you for your insightful feedback regarding the rationale for using interferon-tau (IFN-τ) and lipopolysaccharide (LPS) in our bovine endometrial epithelial cell (bEEC) transcriptomic analysis. We appreciate this opportunity to clarify the scientific basis for our experimental design.

Rationale for IFN-τ and LPS Selection:

(1)LPS is a well-known endotoxin and a major component of the cell wall of Gram-negative bacteria, playing a key role in the pathogenesis of bovine endometritis. Over the years, several scholars have utilized LPS-induced inflammation models in bovine endometrial cells to identify numerous cellular and immune factors involved in the development of bovine endometritis, such as TNF-α, IL-1, IL-6, IL-8, IL-1β, and TLR4. These research findings have laid the foundation for elucidating the pathogenesis of bovine endometritis and for the diagnosis and prevention of the disease.(Line 47-54)

(2) IFN-τ is a type I interferon that is specifically secreted by embryonic trophoblast cells during early pregnancy in ruminants. In addition to acting as a signal for maternal pregnancy recognition in cattle , it also has antiviral, antiproliferative, and immunomodulatory activities . Initially recognized for its anti-luteolytic action through suppression of endometrial PGF2α. IFN-τ is now known to exert systemic effects beyond the uterus. It induces interferon-stimulated genes in endometrial and peripheral immune cells, shaping an immune environment conducive to embryo tolerance. By modulating NF-κB, signal transducer and activator of transcription 1, and interferon regulatory factors, IFN-τ downregulates pro-inflammatory cytokines such as tumor necrosis factor alpha and interferon gamma, while enhancing anti-inflammatory mediators including IL-10 and IL-4. This shift promotes a Thelper 2-dominant immune profile favorable for maternal–fetal tolerance. Previous studies have shown that IFN-τ exerts significant anti-inflammatory effects in inflammatory diseases . Still there are currently few studies exploring IFN-τ in LPS-induced endometritis in cows based on transcriptomics. Transcriptomics analysis is crucial for understanding the occurrence, development, and pathophysiological mechanisms of endometritis in dairy cows, which will facilitate the prevention and treatment of this disease.(Line 64-81)

We sincerely appreciate your meticulous review, which has prompted us to strengthen the methodological justification in the revised manuscript. This clarification significantly enhances the scientific rigor of our work.

4. In the experimental design three groups, Control (C), LPS (L) and IFN� + LPS (F) groups were included. To understand the effect of IFN� on the transcriptomic profile an additional group of IFN� alone is required.

Response: We sincerely apologize for the oversight in our experimental design, which we acknowledge as a critical limitation in the original study. We appreciate your insightful suggestion to include an additional group treated with interferon-tau (IFN-τ) alone, which would indeed strengthen the mechanistic interpretation of the transcriptomic data. The exclusion of an IFN-τ-only group resulted from an initial focus on investigating the interaction between IFN-τ and LPS rather than their independent effects. This prioritization stemmed from our hypothesis that IFN-τ might modulate LPS-induced inflammation during early pregnancy. However, we recognize that this design flaw prevents us from fully dissecting the specific transcriptional contributions of IFN-τ itself. We deeply appreciate your meticulous review, which has highlighted a significant methodological shortfall. Your feedback has prompted us to adopt more rigorous design standards, ultimately improving the scientific validity of our work.

5. Line #198-202: The Q20 base distribution ranged from 98.95% to 99.57% (Q20 > 90% was considered acceptable), Q30 bases ranged from 96.45% to 97.98% (Q30 > 80% was considered acceptable), and GC content ranged from 41.99% to 56.48%. However, the values mentioned in Suppl. Table 1. does not match with the values mentioned in the text. Please check and clarify.

Response: We sincerely apologize for the discrepancy between the quality metrics reported in the main text (Lines 198-202) and those presented in Supplementary Table 1. This inconsistency resulted from an oversight during data compilation, We appreciate your

---

## [Decision Letter · Decision Letter 1]

8 Feb 2026

Exploring the effects of IFN-τ on LPS-induced endometritis in cows based on transcriptomics

PONE-D-25-59265R1

Dear Dr. Liu,

We’re pleased to inform you that your manuscript has been judged scientifically suitable for publication and will be formally accepted for publication once it meets all outstanding technical requirements.

Kind regards,

Vikash Chandra, PhD

Academic Editor

PLOS One

Reviewers' comments:

Reviewer's Responses to Questions

**Comments to the Author**

Reviewer #1: All comments have been addressed

Reviewer #2: All comments have been addressed

2. Is the manuscript technically sound, and do the data support the conclusions?

Reviewer #1: Yes

Reviewer #2: Yes

3. Has the statistical analysis been performed appropriately and rigorously?

Reviewer #1: Yes

Reviewer #2: Yes

4. Have the authors made all data underlying the findings in their manuscript fully available?

Reviewer #1: Yes

Reviewer #2: Yes

5. Is the manuscript presented in an intelligible fashion and written in standard English?

Reviewer #1: Yes

Reviewer #2: Yes

Reviewer #1: Authors have addressed most of the comments raised by me. The revised manuscript is now suitable for acceptance.

Reviewer #2: Unresolved Minor Concern:

1. Corrected the conceptual error about cervical status (revised "during pregnancy" to "during estrus" for pathogen entry in endometritis).

2. Clarified the rationale for IFN-τ/LPS use (LPS as a key endometritis pathogen component; IFN-τ as anti-inflammatory/maternal recognition signal) and added relevant literature on IFN-τ’s role in embryo tolerance.

3. Resolved data inconsistency (Q20/Q30/GC content values now match Suppl. Table 1).

4. Streamlined the conclusion to focus on core findings (IFN-τ’s multi-pathway anti-inflammatory effects) and clinical applications.

5. Improved English grammar via native speaker revision

what does this mean?

---

## [Editor Report · Acceptance letter]

PONE-D-25-59265R1

PLOS One

Dear Dr. Liu,

I'm pleased to inform you that your manuscript has been deemed suitable for publication in PLOS One. Congratulations! Your manuscript is now being handed over to our production team.

Kind regards,

on behalf of

Dr. Vikash Chandra

Academic Editor

PLOS One